# Partition-Based Formulations for Mixed-Integer Optimization of Trained ReLU Neural Networks

**Calvin Tsay**
Department of Computing
Imperial College London
c.tsay@imperial.ac.uk

**Jan Kronqvist**
Department of Mathematics
KTH Royal Institute of Technology
jankr@kth.se

**Alexander Thebelt**
Department of Computing
Imperial College London
alexander.thebelt18@imperial.ac.uk

**Ruth Misener**
Department of Computing
Imperial College London
r.misener@imperial.ac.uk

## Abstract

This paper introduces a class of mixed-integer formulations for trained ReLU neural networks. The approach balances model size and tightness by partitioning node inputs into a number of groups and forming the convex hull over the partitions via disjunctive programming. At one extreme, one partition per input recovers the convex hull of a node, i.e., the tightest possible formulation for each node. For fewer partitions, we develop smaller relaxations that approximate the convex hull, and show that they outperform existing formulations. Specifically, we propose strategies for partitioning variables based on theoretical motivations and validate these strategies using extensive computational experiments. Furthermore, the proposed scheme complements known algorithmic approaches, e.g., optimization-based bound tightening captures dependencies within a partition.

## 1 Introduction

Many applications use mixed-integer linear programming (MILP) to optimize over trained feed-forward ReLU neural networks (NNs) [5, 14, 18, 23, 32, 36]. A MILP encoding of a ReLU-NN enables network properties to be rigorously analyzed, e.g., maximizing a neural acquisition function [33] or verifying robustness of an output (often classification) within a restricted input domain [6]. MILP encodings of ReLU-NNS have also been used to determine robust perturbation bounds [9], compress NNs [28], count linear regions [27], and find adversarial examples [17]. The so-called *big-M* formulation is the main approach for encoding NNs as MILPs in the above references. Optimizing the resulting MILPs remains challenging for large networks, even with state-of-the-art software.

Effectively solving a MILP hinges on the strength of its continuous relaxation [10]; weak relaxations can render MILPs computationally intractable. For NNs, Anderson et al. [3] showed that the big-M formulation is not tight and presented formulations for the convex hull (i.e., the tightest possible formulation) of individual nodes. However, these formulations require either an exponential (*w.r.t.* inputs) number of constraints or many additional/auxiliary variables. So, despite its weaker continuous relaxation, the big-M formulation can be computationally advantageous owing to its smaller size.

Given these challenges, we present a novel class of MILP formulations for ReLU-NNs. The formulations are hierarchical: their relaxations start at a big-M equivalent and converge to the convex hull. Intermediate formulations can closely approximate the convex hull with many fewer variables/constraints. The formulations are constructed by viewing each ReLU node as a two-part

35th Conference on Neural Information Processing Systems (NeurIPS 2021).

disjunction. Kronqvist et al. [21] proposed hierarchical relaxations for general disjunctive programs. This work develops a similar hierarchy to construct strong and efficient MILP formulations specific to ReLU-NNs. In particular, we partition the inputs of each node into groups and formulate the convex hull over the resulting groups. With fewer groups than inputs, this approach results in MILPs that are smaller than convex-hull formulations, yet have stronger relaxations than big-M.

Three optimization tasks evaluate the new formulations: *optimal adversarial examples*, *robust verification*, and $\ell_1$-*minimally distorted adversaries*. Extensive computation, including with convex-hull-based cuts, shows that our formulations outperform the standard big-M approach with 25% more problems solved within a 1h time limit (average 2.2X speedups for solved problems).

**Related work**. Techniques for obtaining strong relaxations of ReLU-NNs include linear programming [16, 34, 35], semidefinite programming [12, 25], Lagrangian decomposition [7], combined relaxations [30], and relaxations over multiple nodes [29]. Recently, Tjandraatmadja et al. [31] derived the tightest possible convex relaxation for a single ReLU node by considering its multivariate input space. These relaxation techniques do not exactly represent ReLU-NNs, but rather derive valid bounds for the network in general. These techniques might fail to verify some properties, due to their non-exactness, but they can be much faster than MILP-based techniques. Strong MILP encoding of ReLU-NNs was also studied by Anderson et al. [3], and a related dual algorithm was later presented by De Palma et al. [13]. Our approach is fundamentally different, as it constructs computationally cheaper formulations that approximate the convex hull, and we start by deriving a stronger relaxation instead of strengthening the relaxation via cutting planes. Furthermore, our formulation enables input node dependencies to easily be incorporated.

**Contributions of this paper.** We present a new class of strong, yet compact, MILP formulations for feed-forward ReLU-NNs in Section 3. Section 3.1 observes how, in conjunction with optimization-based bound tightening, partitioning input variables can efficiently incorporate dependencies into MILP formulations. Section 3.2 builds on the observations of Kronqvist et al. [21] to prove the hierarchical nature of the proposed ReLU-specific formulations, with relaxations spanning between big-M and convex-hull formulations. Sections 3.3–3.4 show that formulation tightness depends on the specific choice of variable partitions, and we present efficient partitioning strategies based on both theoretical and computational motivations. The advantages of the new formulations are demonstrated via extensive computational experiments in Section 4.

## 2 Background

### 2.1 Feed-forward Neural Networks

A feed-forward neural network (NN) is a directed acyclic graph with nodes structured into $k$ layers. Layer $k$ receives the outputs of nodes in the preceding layer $k-1$ as its inputs (layer 0 represents inputs to the NN). Each node in a layer computes a weighted sum of its inputs (known as the *preactivation*), and applies an activation function. This work considers the ReLU activation function:

$$y = \max(0, \boldsymbol{w}^T \boldsymbol{x} + b) \tag{1}$$

where $\boldsymbol{x} \in \mathbb{R}^\eta$ and $y \in [0, \infty)$ are, respectively, the inputs and output of a node ($\boldsymbol{w}^T \boldsymbol{x} + b$ is termed the preactivation). Parameters $\boldsymbol{w} \in \mathbb{R}^\eta$ and $b \in \mathbb{R}$ are the node weights and bias, respectively.

### 2.2 ReLU Optimization Formulations

In contrast to the training of NNs (where parameters $\boldsymbol{w}$ and $b$ are optimized), optimization over a NN seeks extreme cases for a *trained* model. Therefore, model parameters $(\boldsymbol{w}, b)$ are fixed, and the inputs/outputs of nodes in the network $(\boldsymbol{x}, y)$ are optimization variables instead.

**Big-M Formulation.** The ReLU function (1) is commonly modeled [9, 23]:

$$y \geq (\boldsymbol{w}^T \boldsymbol{x} + b) \tag{2}$$

$$y \leq (\boldsymbol{w}^T \boldsymbol{x} + b) - (1 - \sigma)LB^0 \tag{3}$$

$$y \leq \sigma UB^0 \tag{4}$$

where $y \in [0, \infty)$ is the node output and $\sigma \in \{0, 1\}$ is a binary variable corresponding to the on-off state of the neuron. The formulation requires the bounds (big-M coefficients) $LB^0, UB^0 \in \mathbb{R}$, which should be as tight as possible, such that $(\boldsymbol{w}^T\boldsymbol{x} + b) \in [LB^0, UB^0]$.

**Disjunctive Programming** [4]. We observe that (1) can be modeled as a disjunctive program:

$$\begin{bmatrix} y = 0 \\ \boldsymbol{w}^T\boldsymbol{x} + b \leq 0 \end{bmatrix} \vee \begin{bmatrix} y = \boldsymbol{w}^T\boldsymbol{x} + b \\ \boldsymbol{w}^T\boldsymbol{x} + b \geq 0 \end{bmatrix} \tag{5}$$

The extended formulation for disjunctive programs introduces auxiliary variables for each disjunction. Instead of directly applying the standard extended formulation, we first define $z := \boldsymbol{w}^T\boldsymbol{x}$ and assume $z$ is bounded. The auxiliary variables $z^a \in \mathbb{R}$ and $z^b \in \mathbb{R}$ can then be introduced to model (5):

$$\boldsymbol{w}^T\boldsymbol{x} = z^a + z^b \tag{6}$$

$$z^a + \sigma b \leq 0 \tag{7}$$

$$z^b + (1-\sigma)b \geq 0 \tag{8}$$

$$y = z^b + (1-\sigma)b \tag{9}$$

$$\sigma LB^a \leq z^a \leq \sigma UB^a \tag{10}$$

$$(1-\sigma)LB^b \leq z^b \leq (1-\sigma)UB^b \tag{11}$$

where again $y \in [0, \infty)$ and $\sigma \in \{0, 1\}$. Bounds $LB^a, UB^a, LB^b, UB^b \in \mathbb{R}$ are required for this formulation, such that $z^a \in \sigma[LB^a, UB^a]$ and $z^b \in (1-\sigma)[LB^b, UB^b]$. Note that the inequalities in (5) may cause the domains of $z^a$ and $z^b$ to differ. The summation in (6) can be used to eliminate either $z^a$ or $z^b$; therefore, in practice, only one auxiliary variable is introduced by formulation (6)–(11).

**Importance of relaxation strength.** MILP is often solved with branch-and-bound, a strategy that bounds the objective function between the best feasible point and its tightest optimal relaxation. The integral search space is explored by "branching" until the gap between bounds falls below a given tolerance. A *tighter*, or stronger, relaxation can reduce this search tree considerably.

## 3 Disaggregated Disjunctions: Between Big-M and the Convex Hull

Our proposed formulations split the sum $z = \boldsymbol{w}^T\boldsymbol{x}$ into partitions: we will show these formulations have tighter continuous relaxations than (6)–(11). In particular, we partition the set $\{1, ..., \eta\}$ into subsets $\mathbb{S}_1 \cup \mathbb{S}_2 \cup ... \cup \mathbb{S}_N = \{1, ..., \eta\}$; $\mathbb{S}_n \cap \mathbb{S}_{n'} = \emptyset \ \forall n \neq n'$. An auxiliary variable is then introduced for each partition, i.e., $z_n = \sum_{i \in \mathbb{S}_n} w_i x_i$. Replacing $z = \boldsymbol{w}^T\boldsymbol{x}$ with $\sum_{n=1}^{N} z_n$, the disjunction (5) becomes:

$$\begin{bmatrix} y = 0 \\ \sum_{n=1}^{N} z_n + b \leq 0 \end{bmatrix} \vee \begin{bmatrix} y = \sum_{n=1}^{N} z_n + b \\ y \geq 0 \end{bmatrix} \tag{12}$$

Assuming now that each $z_n$ is bounded, the extended formulation can be expressed using auxiliary variables $z_n^a$ and $z_n^b$ for each $z_n$. Eliminating $z_n^a$ via the summation $z_n = z_n^a + z_n^b$ results in our proposed formulation (complete derivation in appendix A.1):

$$\sum_n \left( \sum_{i \in \mathbb{S}_n} w_i x_i - z_n^b \right) + \sigma b \leq 0 \tag{13}$$

$$\sum_n z_n^b + (1-\sigma)b \geq 0 \tag{14}$$

$$y = \sum_n z_n^b + (1-\sigma)b \tag{15}$$

$$\sigma LB_n^a \leq \sum_{i \in \mathbb{S}_n} w_i x_i - z_n^b \leq \sigma UB_n^a, \qquad \forall n = 1, ..., N \tag{16}$$

$$(1-\sigma)LB_n^b \leq z_n^b \leq (1-\sigma)UB_n^b, \qquad \forall n = 1, ..., N \tag{17}$$

where $y \in [0, \infty)$ and $\sigma \in \{0, 1\}$. We observe that (13)–(17) *exactly represents the ReLU node* (given that the partitions $z_n$ are bounded in the original disjunction): the sum $z_n = \sum_{i \in \mathbb{S}_n} w_i x_i$ is

substituted in the extended convex hull formulation [4] of disjunction (12). Note that, compared to the general case presented by Kronqvist et al. [21], both sides of disjunction (12) can be modeled using the same partitions, resulting in fewer auxiliary variables. We further note that domains $[LB_n^a, UB_n^a]$ and $[LB_n^b, UB_n^b]$ may not be equivalent, owing to inequalities in (12).

## 3.1 Obtaining and Tightening Bounds

The big-M formulation (2)–(4) requires valid bounds $(\boldsymbol{w}^T\boldsymbol{x} + b) \in [LB^0, UB^0]$. Given bounds for each input variable, $x_i \in [\underline{x}_i, \bar{x}_i]$, interval arithmetic gives valid bounds:

$$LB^0 = \sum_i \left(\underline{x}_i \max(0, w_i) + \bar{x}_i \min(0, w_i)\right) + b \tag{18}$$

$$UB^0 = \sum_i \left(\bar{x}_i \max(0, w_i) + \underline{x}_i \min(0, w_i)\right) + b \tag{19}$$

But (18)–(19) do not provide the tightest valid bounds in general, as dependencies between the input nodes are ignored. Propagating the resulting over-approximated bounds from layer to layer leads to increasingly large over-approximations, i.e., propagating weak bounds through layers results in a significantly weakened model. These bounds remain in the proposed formulation (13)–(17) in the form of bounds on both the output $y$ and the original variables $\boldsymbol{x}$ (i.e., outputs of the previous layer).

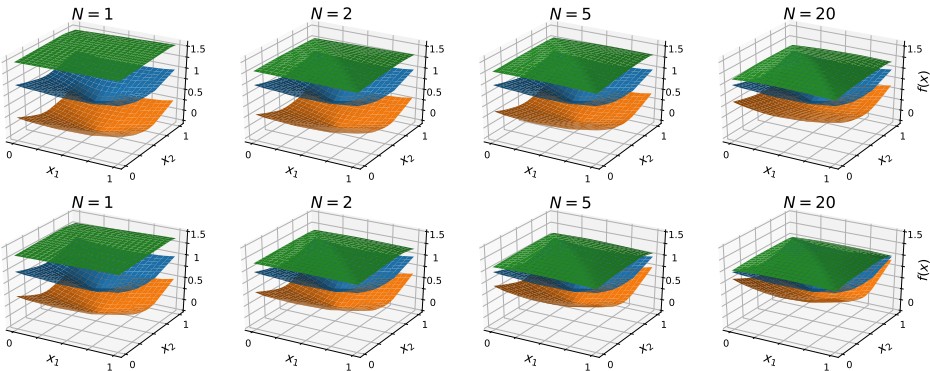

Figure 1: Hierarchical relaxations from $N = 1$ (equiv. big-M) to $N = 20$ (convex hull of each node over a box domain) for a two-input $(x_1, x_2)$ NN trained on scaled Ackley function, with output $f(\boldsymbol{x})$. Top row: $z_n^b$ bounds obtained using interval arithmetic; Bottom row: $z_n^b$ bounds obtained by optimization-based bound tightening. The partitions are formed using the *equal size* strategy.

**Optimization-Based Bound Tightening (OBBT)** or *progressive bounds tightening* [32], tightens variable bounds and constraints [15]. For example, solving the optimization problem with the objective set to minimize/maximize $(\boldsymbol{w}^T\boldsymbol{x} + b)$ gives bounds: $\min(\boldsymbol{w}^T\boldsymbol{x} + b) \leq \boldsymbol{w}^T\boldsymbol{x} + b \leq \max(\boldsymbol{w}^T\boldsymbol{x} + b)$. To simplify these problems, OBBT can be performed using the *relaxed* model (i.e., $\sigma \in [0, 1]$ rather than $\sigma \in \{0, 1\}$), resulting in a linear program (LP). In contrast to (18)–(19), bounds from OBBT incorporate variable dependencies. We apply OBBT by solving one LP per bound.

The partitioned formulation (13)–(17) requires bounds such that $z_n^a \in \sigma[LB_n^a, UB_n^a]$, $z_n^b \in (1 - \sigma)[LB_n^b, UB_n^b]$. In other words, $[LB_n^a, UB_n^a]$ is a valid domain for $z_n^a$ when the node is inactive ($\sigma = 1$) and vice versa. These bounds can also be obtained via OBBT: $\min(\sum_{i \in \mathbb{S}_n} w_i x_i) \leq z_n^a \leq \max(\sum_{i \in \mathbb{S}_n} w_i x_i); \boldsymbol{w}^T\boldsymbol{x} + b \leq 0$. The constraint on the right-hand side of disjunction (12) can be similarly enforced in OBBT problems for $z_n^b$. In our framework, OBBT additionally captures dependencies within each partition $\mathbb{S}_n$. Specifically, we observe that the partitioned OBBT problems effectively form the convex hull over a given polyhedron of the input variables, in contrast to the convex hull formulation, which only considers the box domain defined by the min/max of each input node [3]. Since $\sum_{i \in \mathbb{S}_n} w_i x_i = z_n^a + z_n^b$ and $z_n^a z_n^b = 0$, the bounds $[\min(\sum_{i \in \mathbb{S}_n} w_i x_i), \max(\sum_{i \in \mathbb{S}_n} w_i x_i)]$ are valid for both $z_n^a$ and $z_n^b$. These bounds can be from (18)–(19), or by solving two OBBT problems for each partition ($2N$ LPs total). This simplification uses equivalent bounds for $z_n^a$ and $z_n^b$, but tighter bounds can potentially be found by performing OBBT for $z_n^a$ and $z_n^b$ separately ($4N$ LPs total).

Figure 1 compares the proposed formulation with bounds from interval arithmetic (top row) vs OBBT (bottom row). The true model outputs (blue), and minimum (orange) and maximum (green) outputs of the relaxed model are shown over the input domain. The NNs comprise two inputs, three hidden layers with 20 nodes each, and one output. As expected, OBBT greatly improves relaxation tightness.

## 3.2 Tightness of the Proposed Formulation

**Proposition 1.** (13)–(17) *has the equivalent non-lifted (i.e., without auxiliary variables) formulation:*

$$y \leq \sum_{i \in \mathcal{I}_j} w_i x_i + \sigma(b + \sum_{i \in \mathcal{I} \setminus \mathcal{I}_j} UB_i) + (\sigma - 1)(\sum_{i \in \mathcal{I}_j} LB_i), \qquad \forall j = 1, ..., 2^N \qquad (20)$$

$$y \geq \boldsymbol{w}^T \boldsymbol{x} + b \qquad (21)$$

$$y \leq \sigma UB^0 \qquad (22)$$

*where $UB_i$ and $LB_i$ denote the upper and lower bounds of $w_i x_i$. The set $\mathcal{I}$ denotes the input indices $\{1, ..., \eta\}$, and the subset $\mathcal{I}_j$ denotes the indices contained by the union of the $j$-th combination of partitions in $\{\mathbb{S}_1, ..., \mathbb{S}_N\}$.*

*Proof.* Formulation (13)–(17) introduces $N$ auxiliary variables $z_n^b$, which can be projected out using Fourier-Motzkin elimination (appendix A.2), resulting in combinations $\mathcal{I}_1, ..., \mathcal{I}_J, J = 2^N$. $\qquad \square$

**Remark.** *When $N < \eta$, the family of constraints in (20) represent a subset of the inequalities used to define the convex hull by Anderson et al. [3], where $UB_i$ and $LB_i$ would be obtained using interval arithmetic. Therefore, though a lifted formulation is proposed in this work, the proposed formulations have non-lifted relaxations equivalent to pre-selecting a subset of the convex hull inequalities.*

**Proposition 2.** *Formulation (13)–(17) is equivalent to the big-M formulation (2)–(4) when $N = 1$.*

*Proof.* When $N = 1$, it follows that $\mathbb{S}_1 = \{1, .., \eta\}$. Therefore, $z_1 = \sum_{i=1}^{\eta} w_i x_i = \boldsymbol{w}^T \boldsymbol{x}$, and $\sum_n z_n = z_1 = z$. Conversely, big-M can be seen as the convex hull over a single aggregated "variable," $z = \boldsymbol{w}^T \boldsymbol{x}$. $\qquad \square$

**Proposition 3.** *Formulation (13)–(17) represents the convex hull of (1), given the inputs $\boldsymbol{x}$ are bounded, for the case of $N = \eta$.*

*Proof.* When $N = \eta$, it follows that $\mathbb{S}_n = \{n\}, \forall n = 1, .., \eta$, and, consequently, $z_n = w_n x_n$. Since $z_n$ are linear transformations of each $x_n$ (as are their respective bounds), forming the convex hull over $z_n$ recovers the convex hull over $\boldsymbol{x}$. An extended proof is provided in appendix A.3. $\qquad \square$

**Proposition 4.** *A formulation with $N$ partitions is strictly tighter than any formulation with $(N - 1)$ partitions that is derived by combining two partitions in the former.*

*Proof.* When combining two partitions, i.e., $\mathbb{S}'_{N-1} := \mathbb{S}_{N-1} \cup \mathbb{S}_N$, constraints in (20) where $\mathbb{S}'_{N-1} \subseteq \mathcal{I}_j$ are also obtained by $\{\mathbb{S}_{N-1}, \mathbb{S}_N\} \subseteq \mathcal{I}_j$. In contrast, those obtained by $\mathbb{S}_{N-1} \vee \mathbb{S}_N \subseteq \mathcal{I}_j$ cannot be modeled by $\mathbb{S}'_{N-1}$. Since each constraint in (20) is facet-defining [3] and distinct from each other, omissions result in a less tight formulation. $\qquad \square$

**Remark.** *The convex hull can be formulated with $\eta$ auxiliary variables using (13)–(17) for $N = \eta$, or $2^\eta$ constraints using (20). While these formulations have tighter relaxations, they can perform worse than big-M due to having more difficult branch-and-bound subproblems. Our proposed formulation balances this tradeoff by introducing a hierarchy of relaxations with increasing tightness and size. The convex hull is created over partitions $z_n, n = 1, ..., N$, rather than the input variables $x_n, n = 1, ..., \eta$. Therefore, only $N$ auxiliary variables are introduced, with $N \leq \eta$. The results in this work are obtained using this lifted formulation (13)–(17); Appendix A.5 compares the computational performance of equivalent non-lifted formulations, i.e., involving $2^N$ constraints.*

Figure 1 shows a hierarchy of increasingly tight formulations from $N = 1$ (equiv. big-M ) to $N = 20$ (convex hull of each node over a box input domain). The intermediate formulations approximate the convex-hull ($N = 20$) formulation well, but need fewer auxiliary variables/constraints.

### 3.3 Effect of Input Partitioning Choice

Formulation (13)–(17) creates the convex hull over $z_n = \sum_{i \in \mathbb{S}_n} w_i x_i, n = 1, ..., N$. The hyperparameter $N$ dictates model size, and the choice of subsets, $\mathbb{S}_1, ..., \mathbb{S}_N$, can strongly impact the strength of its relaxation. By Proposition 4, (13)–(17) can in fact give multiple hierarchies of formulations. While all hierarchies eventually converge to the convex hull, we are primarily interested in those with tight relaxations for small $N$.

**Bounds and Bounds Tightening.** Bounds on the partitions play a key role in the proposed formulation. For example, consider when a node is inactive: $\sigma = 1$, $z_n^b = 0$, and (16) gives the bounds $\sigma LB_n^a \leq \sum_{i \in \mathbb{S}_n} w_i x_i \leq \sigma UB_n^a$. Intuitively, the proposed formulation represents the convex hull over the auxiliary variables, $z_n = \sum_{i \in \mathbb{S}_n} w_i x_i$, and their bounds play a key role in model tightness. We hypothesize these bounds are most effective when partitions $\mathbb{S}_n$ are selected such that $w_i x_i, \forall i \in \mathbb{S}_n$ are of similar orders of magnitude. Consider for instance the case of $\boldsymbol{w} = [1, 1, 100, 100]$ and $x_i \in [0, 1], i = 1, ..., 4$. As all weights are positive, interval arithmetic gives $0 \leq \sum x_i w_i \leq \sum \bar{x}_i w_i$. With two partitions, the choices of $\mathbb{S}_1 = \{1, 2\}$ vs $\mathbb{S}_1 = \{1, 3\}$ give:

$$\begin{bmatrix} x_1 + x_2 \leq \sigma 2 \\ x_3 + x_4 \leq \sigma 2 \end{bmatrix} \text{vs.} \begin{bmatrix} x_1 + 100x_3 \leq \sigma 101 \\ x_2 + 100x_4 \leq \sigma 101 \end{bmatrix} \tag{23}$$

where $\sigma$ is a binary variable. The constraints on the right closely approximate the $\eta$-partition (i.e., convex hull) bounds: $x_3 \leq \sigma$ and $x_4 \leq \sigma$. But $x_1$ and $x_2$ are relatively unaffected by a perturbation $\sigma = 1 - \delta$ (when $\sigma$ is relaxed). Whereas the formulation on the left constrains the four variables equivalently. If the behavior of the node is dominated by a few inputs, the formulation on the right is strong, as it approximates the convex hull over those inputs ($z_1 \approx x_3$ and $z_2 \approx x_4$ in this case). For the practical case of $N << \eta$, there are likely fewer partitions than dominating variables.

The size of the partitions can also be selected to be uneven:

$$\begin{bmatrix} x_1 \leq \sigma 1 \\ x_2 + 100x_3 + 100x_4 \leq \sigma 201 \end{bmatrix} \text{vs.} \begin{bmatrix} x_3 \leq \sigma 1 \\ x_1 + x_2 + 100x_4 \leq \sigma 102 \end{bmatrix} \tag{24}$$

Similar tradeoffs are seen here: the first formulation provides the tightest bound for $x_1$, but $x_2$ is effectively unconstrained and $x_3, x_4$ approach "equal treatment." The second formulation provides the tightest bound for $x_3$, and a tight bound for $x_4$, but $x_1, x_2$ are effectively unbounded for fractional $\sigma$. The above analyses also apply to the case of OBBT. For the above example, solving a relaxed model for $\max(x_1 + x_2)$ obtains a bound that affects the two variables equivalently, while the same procedure for $\max(x_1 + 100x_3)$ obtains a bound that is much stronger for $x_3$ than for $x_1$. Similarly, $\max(x_1 + x_2)$ captures dependency between the two variables, while $\max(x_1 + 100x_3) \approx \max(100x_3)$.

**Relaxation Tightness.** The partitions (and their bounds) also directly affect the tightness of the relaxation for the output variable $y$. The equivalent non-lifted realization (20) reveals the tightness of the above simple example. With two partitions, the choices of $\mathbb{S}_1 = \{1, 2\}$ vs $\mathbb{S}_1 = \{1, 3\}$ result in the equivalent non-lifted constraints:

$$\begin{bmatrix} y \leq x_1 + 100x_3 + \sigma(b + 101) \\ y \leq x_2 + 100x_4 + \sigma(b + 101) \end{bmatrix} \text{vs.} \begin{bmatrix} y \leq x_1 + x_2 + \sigma(b + 200) \\ y \leq 100x_3 + 100x_4 + \sigma(b + 2) \end{bmatrix} \tag{25}$$

Note that combinations $\mathcal{I}_j = \emptyset$ and $\mathcal{I}_j = \{1, 2, 3, 4\}$ in (20) are not analyzed here, as they correspond to the big-M/1-partition model and are unaffected by choice of partitions. The 4-partition model is the tightest formulation and (in addition to all possible 2-partition constraints) includes:

$$y \leq x_i + \sigma(b + 201), i = 1, 2 \tag{26}$$
$$y \leq 100x_i + \sigma(b + 102), i = 3, 4 \tag{27}$$

The unbalanced 2-partition, i.e., the left of (25), closely approximates two of the 4-partition (i.e., convex hull) constraints (27). Analogous to the tradeoffs in terms of bound tightness, we see that this formulation essentially neglects the dependence of $y$ on $x_1, x_2$ and instead creates the convex hull over $z_1 = x_1 + 100x_3 \approx x_3$ and $z_2 \approx x_4$. For this simple example, the behavior of $y$ is dominated by $x_3$ and $x_4$, and this turns out to be a relatively strong formulation. However, when $N << \eta$, we expect neglecting the dependence of $y$ on some input variables to weaken the model.

The alternative formulation, i.e., the right of (25), handles the four variables similarly and creates the convex hull in two new directions: $z_1 = x_1 + x_2$ and $z_2 = 100(x_3 + x_4)$. While this does not

model the individual effect of either $x_3$ or $x_4$ on $y$ as closely, it includes dependency between $x_3$ and $x_4$. Furthermore, $x_1$ and $x_2$ are modeled equivalently (i.e., less tightly than individual constraints). Analyzing partitions with unequal size reveals similar tradeoffs. This section shows that the proposed formulation has a relaxation equivalent to selecting a subset of constraints from (20), with a tradeoff between modeling the effect of individual variables well vs the effect of many variables weakly.

**Deriving Cutting Planes from Convex Hull.** The convex-hull constraints (20) not implicitly modeled by a partition-based formulation can be viewed as potential tightening constraints, or cuts. Anderson et al. [3] provide a linear-time method for selecting the most violated of the exponentially many constraints (20), which is naturally compatible with our proposed formulation (some constraints will be always satisfied). Note that the computational expense can still be significant for practical instances; Botoeva et al. [5] found adding cuts to <0.025% of branch-and-bound nodes to balance their expense and tightening, while De Palma et al. [13] proposed adding cuts at the root node only.

**Remark.** *While the above 4-D case may seem to favor "unbalanced" partitions, it is difficult to illustrate the case where $N << \eta$. Our experiments confirm "balanced" partitions perform better.*

### 3.4   Strategies for Selecting Partitions

The above rationale motivates selecting partitions that result in a model that treats input variables (approximately) equivalently for the practical case of $N << \eta$. Specifically, we seek to evenly distribute tradeoffs in model tightness among inputs. Section 3.3 suggests a reasonable approach is to select partitions such that weights in each are approximately equal (weights are fixed during optimization). We propose two such strategies below, as well as two strategies in **red** for comparison.

**Equal-Size Partitions.** One strategy is to create partitions of *equal size*, i.e., $|\mathbb{S}_1| = |\mathbb{S}_2| = ... = |\mathbb{S}_N|$ (note that they may differ by up to one if $\eta$ is not a multiple of $N$). Indices are then assigned to partitions to keep the weights in each as close as possible. This is accomplished by sorting the weights $\boldsymbol{w}$ and distributing them evenly among partitions (`array_split(argsort(`$\boldsymbol{w}$`)`, $N$) in Numpy).

**Equal-Range Partitions.** A second strategy is to partition with *equal range*, i.e., $\text{range}(w_i) = ... = \text{range}(w_i)$. We define thresholds $\boldsymbol{v} \in \mathbb{R}^{N+1}$ such that $v_1$ and $v_{N+1}$ are $\min(\boldsymbol{w})$ and $\max(\boldsymbol{w})$. To reduce the effect of outliers, we define $v_2$ and $v_N$ as the 0.05 and 0.95 quantiles of $\boldsymbol{w}$, respectively. The remaining elements of $\boldsymbol{v}$ are distributed evenly in $(v_2, v_N)$. Indices $i \in \{1, ..., \eta\}$ are then assigned to $\mathbb{S}_n : w_i \in [v_n, v_{n+1})$. This strategy requires $N \geq 3$, but, for a symmetrically distributed weight vector, $\boldsymbol{w}$, two partitions of equal size are also of equal range.

We compare our proposed strategies against the following:

**Random Partitions.** Input indices $\{1, ..., \eta\}$ are assigned randomly to partitions $\mathbb{S}_1, ..., \mathbb{S}_N$.

**Uneven Magnitudes.** Weights are sorted and "dealt" in reverse to partitions in snake-draft order.

## 4   Experiments

Computational experiments were performed using Gurobi v 9.1 [19], and models were implemented and trained using PyTorch [24]. The computational set-up, solver settings, and models for MNIST [22] and CIFAR-10 [20] are described in appendix A.4.

### 4.1   Optimal Adversary Results

The *optimal adversary* problem takes a target image $\bar{\boldsymbol{x}}$, its correct label $j$, and an adversarial label $k$, and finds the image within a range of perturbations maximizing the difference in predictions. Mathematically, this is formulated as $\max_{\boldsymbol{x}}(f_k(\boldsymbol{x}) - f_j(\boldsymbol{x})); x \in \mathcal{X}$, where $f_k$ and $f_j$ correspond to the $k$- and $j$-th elements of the NN output layer, respectively, and $\mathcal{X}$ defines the domain of perturbations. We consider perturbations defined by the $\ell_1$-norm ($||\boldsymbol{x} - \bar{\boldsymbol{x}}||_1 \leq \epsilon_1 \in \mathbb{R}$), which promotes sparse perturbations [8]. For each dataset, we use the first 100 test-dataset images and randomly selected adversarial labels (the same 100 instances are used for models trained on the same dataset).

Table 1 gives the optimal adversary results. Perturbations $\epsilon_1$ were selected such that some big-M problems were solvable within 3600s (problems become more difficult as $\epsilon_1$ increases). While our

Table 1: Number of solved (in 3600s) optimal adversary problems and average solve times for big-M vs $N = \{2, 4\}$ equal-size partitions. Average times computed for problems solved by all 3 formulations. Grey indicates partition formulations strictly outperforming big-M.

| Dataset | Model | $\epsilon_1$ | Big-M | | 2 Partitions | | 4 Partitions | |
|---|---|---|---|---|---|---|---|---|
| | | | solved | avg.time(s) | solved | avg.time(s) | solved | avg.time(s) |
| MNIST | $2 \times 50$ | 5 | 100 | 57.6 | 100 | **42.7** | 100 | 83.9 |
| | $2 \times 50$ | 10 | 97 | 431.8 | 98 | **270.4** | 98 | 398.4 |
| | $2 \times 100$ | 2.5 | 92 | 525.2 | 100 | **285.1** | 94 | 553.9 |
| | $2 \times 100$ | 5 | 38 | 1587.4 | 59 | **587.8** | 48 | 1084.7 |
| | CNN1* | 0.25 | 68 | 1099.7 | 86 | **618.8** | 87 | 840.0 |
| | CNN1* | 0.5 | 2 | 2293.2 | 16 | **1076.0** | 11 | 2161.2 |
| CIFAR-10 | $2 \times 100$ | 5 | 62 | 1982.3 | 69 | 1083.4 | 85 | **752.8** |
| | $2 \times 100$ | 10 | 23 | 2319.0 | 28 | 1320.2 | 34 | **1318.1** |

*OBBT performed on all NN nodes

formulations consistently outperform big-M in terms of problems solved and solution times, the best choice of $N$ is problem-dependent. For instance, the 4-partition formulation performs best for the $2 \times 100$ network trained on CIFAR-10; Figure 2 (right) shows the number of problems solved for this case. The 2-partition formulation is best for easy problems, but is soon overtaken by larger values of $N$. Intuitively, simpler problems are solved with fewer branch-and-bound nodes and benefit more from smaller subproblems. Performance declines again near $N \geq 7$, supporting observations that the convex-hull formulation ($N = \eta$) is not always best [3].

**Convex-Hull-Based Cuts.** The cut-generation strategy for big-M by Anderson et al. [3] is compatible with our formulations (Section 3.3), i.e., we begin with a partition-based formulation and apply cuts during optimization. Figure 2 (left) gives results with these cuts added (via Gurobi callbacks) at various cut frequencies. Specifically, a cut frequency of $1/k$ corresponds to adding the most violated cut (if any) to each NN node at every $k$ branch-and-bound nodes. Performance is improved at certain frequencies; however, our formulations consistently outperform even the best cases using big-M with cuts. Moreover, cut generation is not always straightforward to integrate with off-the-shelf MILP solvers, and our formulations often perform just as well (if not better) without added cuts. Appendix A.5 gives results with most-violated cuts directly added to the model, as in De Palma et al. [13].

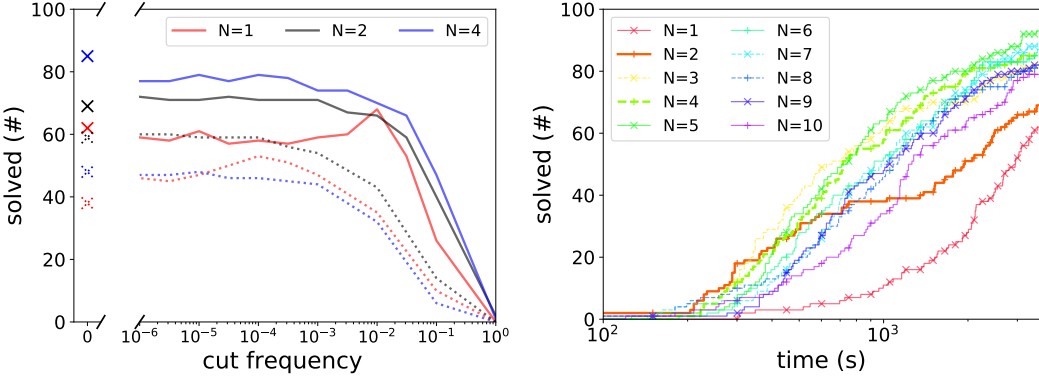

Figure 2: Number solved vs convex-hull-cut frequency/run time for optimal adversary problems ($\epsilon = 5$) for $2 \times 100$ models, without OBBT. Left: $N = 1$ (equivalent to big-M) performs worst on MNIST (dotted) and CIFAR-10 (solid) models for most cut frequencies. Right: each line shows 100 runs of CIFAR-10 model (no convex-hull cuts). $N = 1$ performs the worst; $N = 2$ performs well for easier problems; and intermediate values of $N$ balance model size and tightness well.

**Partitioning Strategies.** Figure 3 shows the result of the input partitioning strategies from Section 3.4 on the MNIST $2 \times 100$ model for varying $N$. Both proposed strategies (blue) outperform formulations with random and uneven partitions (red). With OBBT, big-M ($N = 1$) outperforms partition-based formulations when partitions are selected poorly. Figure 3 again shows that our formulations perform best for some intermediate $N$; random/uneven partitions worsen with increasing $N$.

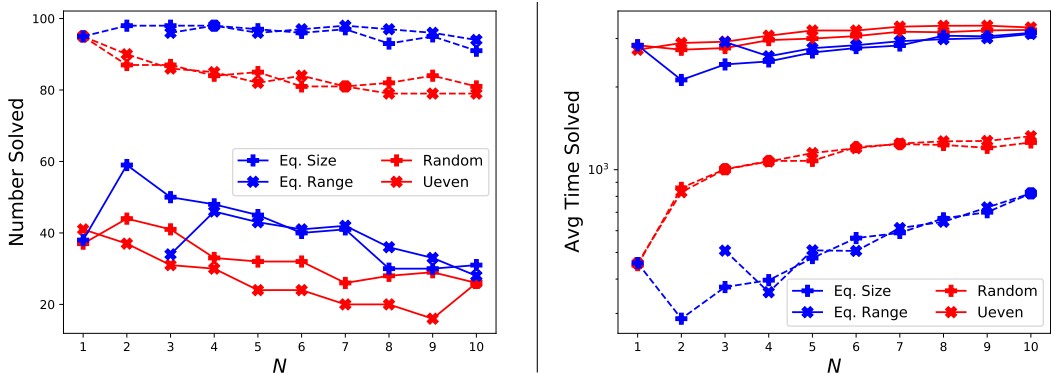

Figure 3: Number solved (left) and solution times (right) for optimal adversary problems for MNIST $2 \times 100$ ($\epsilon = 5$). Each point shows 100 runs, max time of 3600s. Dashed lines show runs with OBBT. The equal range strategy requires $N \geq 3$. Our proposed (blue) partitioning strategies solve more problems (top) faster (bottom) than random and uneven partitions (red)

**Optimization-Based Bounds Tightening.** OBBT was implemented by tightening bounds for all $z_n^b$. We found that adding the bounds from the 1-partition model (i.e., bounds on $\sum z_n^b$) improved all models, as they account for dependencies among all inputs. Therefore these bounds were used in all models, resulting in $2N + 2$ LPs per node ($N \geq 2$). We limited the OBBT LPs to 5s; interval bounds were used if an LP was not solved. Figure 3 shows that OBBT greatly improves the optimization performance of all formulations. OBBT problems for each layer are independent and could, in practice, be solved in parallel. Therefore, at a minimum, OBBT requires the sum of max solution times found in each layer (5s × # layers in this case). This represents an avenue to significantly improve MILP optimization of NNs via parallelization. In contrast, parallelizing branch-and-bound is known to be challenging and may have limited benefits [1, 26].

Table 2: Number of verification problems solved in 3600s and average solve times for big-M vs $N = \{2, 4\}$ equal-size partitions. Average times computed for problems solved by all 3 formulations. OBBT was performed for all problems. Grey indicates formulations strictly outperforming big-M.

| Dataset | Model | $\epsilon_{\infty}$ | Big-M | | 2 Partitions | | 4 Partitions | |
|---|---|---|---|---|---|---|---|---|
| | | | sol.(#) | avg.time(s) | sol.(#) | avg.time(s) | sol.(#) | avg.time(s) |
| MNIST | CNN1 | 0.050 | 82 | 198.5 | 92 | **27.3** | 90 | 52.4 |
| | CNN1 | 0.075 | 30 | 632.5 | 52 | **139.6** | 42 | 281.6 |
| | CNN2 | 0.075 | 21 | 667.1 | 36 | **160.7** | 31 | 306.0 |
| | CNN2 | 0.100 | 1 | 505.3 | 5 | **134.7** | 5 | 246.3 |
| CIFAR-10 | CNN1 | 0.007 | 99 | 100.6 | 100 | **25.9** | 99 | 45.4 |
| | CNN1 | 0.010 | 98 | 119.1 | 100 | **25.7** | 100 | 45.0 |
| | CNN2 | 0.007 | 80 | 300.5 | 95 | **85.1** | 68 | 928.6 |
| | CNN2 | 0.010 | 40 | 743.6 | 72 | **176.4** | 35 | 1041.3 |

## 4.2 Verification Results

The *verification* problem is similar to the optimal adversary problem, but terminates when the sign of the objective function is known (i.e., the lower/upper bounds of the MILP have the same sign). This problem is typically solved for perturbations defined by the $\ell_{\infty}$-norm ($||\boldsymbol{x} - \bar{\boldsymbol{x}}||_{\infty} \leq \epsilon_{\infty} \in \mathbb{R}$). Here, problems are difficult for moderate $\epsilon_{\infty}$: at large $\epsilon_{\infty}$ a mis-classified example (positive objective) is easily found. Several verification tools rely on an underlying big-M formulation, e.g., MIPVerify [32], NSVerify [2], making big-M an especially relevant point of comparison. Owing to the early termination, larger NNs can be tested compared to the optimal adversary problems. We turned off cuts (`cuts= 0`) for the partition-based formulations, as the models are relatively tight over the box-domain perturbations and do not seem to benefit from additional cuts. On the other hand, removing cuts improved some problems using big-M and worsened others. Results are presented in Table 2. Our formulations again generally outperform big-M ($N$=1), except for a few of the 4-partition problems.

Table 3: Number of $\ell_1$-minimally distorted adversary problems solved in 3600s and average solve times for big-M vs $N = \{2, 4\}$ equal-size partitions. Average times and $\epsilon_1$ are computed for problems solved by all 3 formulations. OBBT was performed for all problems. Grey indicates partition formulations strictly outperforming big-M.

| Dataset | Model | avg($\epsilon_1$) | Big-M | | 2 Partitions | | 4 Partitions | |
|---------|-------|---------|--------|------------|--------|------------|--------|------------|
| | | | solved | avg.time(s) | solved | avg.time(s) | solved | avg.time(s) |
| MNIST | $2 \times 50$ | 6.51 | 52 | 675.0 | 93 | **150.9** | 89 | 166.6 |
| | $2 \times 75$ | 4.41 | 16 | 547.3 | 37 | **310.5** | 31 | 424.0 |
| | $2 \times 100$ | 2.73 | 7 | 710.8 | 13 | **572.9** | 10 | 777.9 |

### 4.3 Minimally Distorted Adversary Results

In a similar vein as Croce and Hein [11], we define the $\ell_1$-**minimally distorted adversary** problem: given a target image $\bar{x}$ and its correct label $j$, find the smallest perturbation over which the NN predicts an adversarial label $k$. We formulate this as $\min_{\epsilon_1, x} \epsilon_1; ||x - \bar{x}||_1 \leq \epsilon_1; f_k(x) \geq f_j(x)$. The adversarial label $k$ is selected as the second-likeliest class of the target image. Figure 4 provides examples illustrating that the $\ell_1$-norm promotes sparse perturbations, unlike the $\ell_\infty$-norm. Note that the sizes of perturbations $\epsilon_1$ are dependent on input dimensionality; if it were distributed evenly, $\epsilon_1 = 5$ would correspond to an $\ell_\infty$-norm perturbation $\epsilon_\infty \approx 0.006$ for MNIST models.

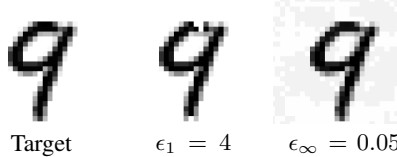

Target        $\epsilon_1 = 4$        $\epsilon_\infty = 0.05$

Figure 4: Sample $\ell_1$- vs $\ell_\infty$-based minimally distorted adversaries for the MNIST $2 \times 50$ model. The true label ($j$) is '9,' and the adversarial label ($k$) is '4.'

Table 3 presents results for the minimally distorted adversary problems. As input domains are unbounded, these problems are considerably more difficult than the above *optimal adversary* problems. Therefore, only smaller MNIST networks were manageable (with OBBT) for all formulations. Again, partition-based formulations consistently outperform big-M, solving more problems and in less time.

## 5 Conclusions

This work presented MILP formulations for ReLU NNs that balance having a tight relaxation and manageable size. The approach forms the convex hull over partitions of node inputs; we presented theoretical and computational motivations for obtaining good partitions for ReLU nodes. Furthermore, our approach expands the benefits of OBBT, which, unlike conventional MILP tools, can easily be parallelized. Results on three classes of optimization tasks show that the proposed formulations consistently outperform standard MILP encodings, allowing us to solve 25% more of the problems (average >2X speedup for solved problems). Implementations of the proposed formulations and partitioning strategies are available at `https://github.com/cog-imperial/PartitionedFormulations_NN`.

### Acknowledgments

This work was supported by Engineering & Physical Sciences Research Council (EPSRC) Fellowships to CT and RM (grants EP/T001577/1 and EP/P016871/1), an Imperial College Research Fellowship to CT, a Royal Society Newton International Fellowship (NIF\R1\182194) to JK, a grant by the Swedish Cultural Foundation in Finland to JK, and a PhD studentship funded by BASF to AT.

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
