$ in the extended convex hull formulation [4] of the original disjunction (5), directly gives (29)–(34). Eliminating $z_n^a$ via (29) results in our proposed formulation:

$$\sum_n \left( \sum_{i \in \mathbb{S}_n} w_i x_i - z_n^b \right) + \sigma b \leq 0 \tag{35}$$

$$\sum_n z_n^b + (1 - \sigma)b \geq 0 \tag{36}$$

$$y = \sum_n z_n^b + (1 - \sigma)b \tag{37}$$

$$\sigma LB_n^a \leq \sum_{i \in \mathbb{S}_n} w_i x_i - z_n^b \leq \sigma UB_n^a, \qquad\qquad \forall n = 1, ..., N \tag{38}$$

$$(1 - \sigma)LB_n^b \leq z_n^b \leq (1 - \sigma)UB_n^b, \qquad\qquad \forall n = 1, ..., N \tag{39}$$

## A.2   Derivation of Equivalent Non-Extended Formulation

We seek an equivalent (non-lifted) formulation to (35)–(39) without $z_n^b$. The term $\sum_n z_n^b$ can be isolated in (35)–(37), giving:

$$\sum_n z_n^b \geq \boldsymbol{w}^T \boldsymbol{x} + \sigma b \tag{40}$$

$$\sum_n z_n^b \geq -(1 - \sigma)b \tag{41}$$

$$\sum_n z_n^b = y - (1 - \sigma)b \tag{42}$$

We observed in Section 3.1 that $[\min(\sum_{i \in \mathbb{S}_n} w_i x_i), \max(\sum_{i \in \mathbb{S}_n} w_i x_i)]$ are valid for both $z_n^a$ and $z_n^b$. Let $UB_i$ and $LB_i$ denote, respectively, valid upper and lower bounds of weighted input $w_i x_i$.

Isolating $z_n^b$ in the bounds (38)–(39) gives:

$$z_n^b \leq \sum_{i \in \mathbb{S}_n} w_i x_i - \sigma \sum_{i \in \mathbb{S}_n} LB_i \tag{43}$$

$$z_n^b \geq \sum_{i \in \mathbb{S}_n} w_i x_i - \sigma \sum_{i \in \mathbb{S}_n} UB_i \tag{44}$$

$$z_n^b \leq (1 - \sigma) \sum_{i \in \mathbb{S}_n} UB_i \tag{45}$$

$$z_n^b \geq (1 - \sigma) \sum_{i \in \mathbb{S}_n} LB_i \tag{46}$$

**Fourier-Motzkin Elimination.** We will now eliminate the auxiliary variables using the above inequalities. Equations that appear in the non-extended formulation are marked in green. First, we examine the equations containing $\sum_n z_n^b$. Combining (40)+(42) and (41)+(42) gives, respectively:

$$\textcolor{green}{y \geq \boldsymbol{w}^T \boldsymbol{x} + b} \tag{47}$$

$$\textcolor{green}{y \geq 0} \tag{48}$$

Secondly, we examine equations containing only $z_n^b$. Combining (43)+(44) gives the trivial constraint $\sum_{i \in \mathbb{S}_n} UB_i \geq \sum_{i \in \mathbb{S}_n} LB_i$. Combining (45)+(46) gives the same constraint. The other combinations (43)+(46) and (44)+(45) recover the definition of lower and upper bounds on the partitions:

$$\sum_{i \in \mathbb{S}_n} LB_i \leq \sum_{i \in \mathbb{S}_n} w_i x_i \leq \sum_{i \in \mathbb{S}_n} UB_i \tag{49}$$

Now examining both equations containing $z_n^b$ and those containing $\sum_n z_n^b$, the combinations between (42) and (43)–(46) are most interesting. Here, each $z_n^b$ can be eliminated using either inequality (43) or (45). Note that combining these with (40) or (41) results in trivially redundant constraints. Defining $\mathcal{A}$ as the indices of partitions for which (43) is used, and $\mathcal{B}$ as the remaining indices:

$$y - (1 - \sigma)b \leq \sum_{n \in \mathcal{A}} \sum_{i \in \mathcal{S}_n} w_i x_i - \sigma \sum_{n \in \mathcal{A}} \sum_{i \in \mathcal{S}_n} LB_i + (1 - \sigma) \sum_{n \in \mathcal{B}} \sum_{i \in \mathcal{S}_n} UB_i \tag{50}$$

Similarly, the inequalities of opposite sign can be chosen from either (44) or (46). Defining now $\mathcal{A}'$ as the indices of partitions for which (44) is used, and $\mathcal{B}'$ as the remaining indices:

$$y - (1 - \sigma)b \geq \sum_{n \in \mathcal{A}'} \sum_{i \in \mathcal{S}_n} w_i x_i - \sigma \sum_{n \in \mathcal{A}'} \sum_{i \in \mathcal{S}_n} UB_i + (1 - \sigma) \sum_{n \in \mathcal{B}} \sum_{i \in \mathcal{S}_n} LB_i \tag{51}$$

Defining $\mathcal{I}_j$ as the union of $\mathbb{S}_n \forall n \in \mathcal{A}$ and $\mathcal{I}_j'$ as the union of $\mathbb{S}_n \forall n \in \mathcal{A}'$, (50)–(51) become:

$$y \leq \sum_{i \in \mathcal{I}_j} w_i x_i - \sigma \sum_{i \in \mathcal{I}_j} LB_i + (1 - \sigma)(b + \sum_{i \in \mathcal{I} \setminus \mathcal{I}_j} UB_i) \tag{52}$$

$$y \geq \sum_{i \in \mathcal{I}_j'} w_i x_i - \sigma \sum_{i \in \mathcal{I}_j'} UB_i + (1 - \sigma)(b + \sum_{i \in \mathcal{I} \setminus \mathcal{I}_j'} LB_i) \tag{53}$$

The lower inequality (53) is redundant. Consider that $\sigma \in \{0, 1\}$. For $\sigma = 0$ and $\sigma = 1$, we recover:

$$y \geq \sum_{i \in \mathcal{I}_j'} w_i x_i + (b + \sum_{i \in \mathcal{I} \setminus \mathcal{I}_j'} LB_i) \tag{54}$$

$$y \geq \sum_{i \in \mathcal{I}_j'} (w_i x_i - UB_i) \tag{55}$$

The former is less tight than $y \geq \boldsymbol{w}^T \boldsymbol{x} + b$, while the latter is less tight than $y \geq 0$. Finally, setting $\sigma' = 1 - \sigma$ in (52) gives:

$$\textcolor{green}{y \leq \sum_{i \in \mathcal{I}_j} w_i x_i + (\sigma' - 1) \sum_{i \in \mathcal{I}_j} LB_i + \sigma'(b + \sum_{i \in \mathcal{I} \setminus \mathcal{I}_j} UB_i)} \tag{56}$$

The combination $\mathcal{I}_j = \emptyset$ removes all $x_i$, giving an upper bound for $y$:

$$y \leq \sigma'(b + \sum_{i \in \mathcal{I}} UB_i) \tag{57}$$

Combining all remaining equations gives a nonextended formulation:

$$y \leq \sum_{i \in \mathcal{I}_j} w_i x_i + \sigma'(b + \sum_{i \in \mathcal{I} \setminus \mathcal{I}_j} UB_i) + (\sigma' - 1)(\sum_{i \in \mathcal{I}_j} LB_i) \tag{58}$$

$$y \geq \boldsymbol{w}^T \boldsymbol{x} + b \tag{59}$$

$$y \leq \sigma' UB^0 \tag{60}$$

$$y \in [0, \infty) \tag{61}$$

### A.3 Equivalence of $\eta$-Partition Formulation to Convex Hull

When $N = \eta$, it follows that $\mathbb{S}_n = \{n\}, \forall n = 1, .., \eta$, and, consequently, $z_n = w_n x_n$. The auxiliary variables can be expressed in terms of $x_n$, rather of $z_n$ (i.e., $z_n^a = w_n x_n^a$ and $z_n^b = w_n x_n^b$). Rewriting (29)–(34) in this way gives:

$$x_n = x_n^a + x_n^b \tag{62}$$

$$\boldsymbol{w}^T \boldsymbol{x}^a + \sigma b \leq 0 \tag{63}$$

$$\boldsymbol{w}^T \boldsymbol{x}^b + (1 - \sigma)b \geq 0 \tag{64}$$

$$y = \boldsymbol{w}^T \boldsymbol{x}^b + (1 - \sigma)b \tag{65}$$

$$\sigma \frac{LB_n^a}{w_n} \leq x_n^a \leq \sigma \frac{UB_n^a}{w_n}, \qquad \forall n = 1, ..., \eta \tag{66}$$

$$(1 - \sigma) \frac{LB_n^b}{w_n} \leq x_n^b \leq (1 - \sigma) \frac{UB_n^b}{w_n}, \qquad \forall n = 1, ..., \eta \tag{67}$$

This formulation with a copy of each input—sometimes referred to as a "multiple choice" formulation—represents the convex hull of the node, e.g., see [3]; however, the overall model tightness still hinges on the bounds used for $x_n, n = 1, ..., \eta$. We note that (66)–(67) are presented here with $x_n^a$ and $x_n^b$ isolated for clarity. In practice, the bounds can be written without $w_n$ in their denominators, as in (13)–(17).

### A.4 Experiment Details

**Computational Set-Up.** All computational experiments were performed on a 3.2 GHz Intel Core i7-8700 CPU (12 threads) with 16 GB memory. Models were implemented and solved using Gurobi v 9.1 [19] (academic license). The LP algorithm was set to dual simplex, `cuts = 1` (moderate cut generation), `TimeLimit = 3600s`, and default termination criteria were used. We set parameter `MIPFocus = 3` to ensure consistent solution approaches across formulations. Average times were computed as the arithmetic mean of solve times for instances solved by all formulations for a particular problem class. Thus, no time outs are included in the calculation of average solve times.

**Neural Networks.** We trained several NNs on MNIST [22] and CIFAR-10 [20], including both fully connected NNs and convolutional NNs (CNNs). MNIST (CC BY-SA 3.0) comprises a training set of 60,000 images and a test set of 10,000 images. CIFAR-10 (MIT License) comprises a training set of 50,000 images and a test set of 10,000 images. Dense models are denoted by $n_{\text{Layers}} \times n_{\text{Hidden}}$ and comprise $n_{\text{Layers}} \times n_{\text{Hidden}}$ hidden plus 10 output nodes. CNN2 is based on 'ConvSmall' of the ERAN dataset [29]: {`Conv2D(16, (4,4), (2,2))`, `Conv2D(32, (4,4), (2,2))`, `Dense(100)`, `Dense(10)`}. CNN1 halves the channels in each convolutional layer: {`Conv2D(8, (4,4), (2,2))`, `Conv2D(16, (4,4), (2,2))`, `Dense(100)`, `Dense(10)`}. The implementations of CNN1/CNN2 have 1,852/3,604 and 2,476/4,852 nodes for MNIST and CIFAR-10, respectively. NNs are implemented in PyTorch [24] and obtained using standard training (i.e., without regularization or methods to improve robustness). MNIST models were trained for ten epochs, and CIFAR-10 models were trained for 20 epochs, all using the `Adadelta` optimizer with default parameters.

### A.5 Additional Computational Results

**Comparing formulations.** Proposition 1 suggests an equivalent, non-lifted formulation for each partition-based formulation. In contrast to our proposed formulation (13)–(17), which adds a linear (*w.r.t.* number of partitions $N$) number of variables, this *non-lifted* formulation involves adding an exponential (*w.r.t.* $N$) number of constraints. Table 4 computationally compares our proposed formulations against non-lifted formulations with equivalent relaxations on 100 optimal adversary instances of medium difficulty. These results clearly show the advantages of the proposed formulations; a non-lifted formulation equivalent to 20-partitions ($2^{20}$ constraints per node) cannot even be fully generated on our test machines. Additionally, our formulations naturally enable OBBT on partitions. As described in Section 3.1, this captures dependencies among each partition without additional constraints.

Table 4: Number of solved (in 3600s) optimal adversary problems for $N = \{2, 5, 10, 20\}$ equal-size partitions for MNIST $2 \times 100$ model, $\epsilon_1 = 5$. Performance is compared between proposed formulations and non-lifted formulations with equivalent relaxations.

| Partitions | Proposed Formulation | | Non-Lifted Formulation | |
| $N$ | solved | avg.time(s) | solved | avg.time(s) |
| --- | --- | --- | --- | --- |
| 2 | 59 | 530.1 | 41 | 1111.4 |
| 5 | 45 | 792.5 | 7 | 2329.8 |
| 10 | 31 | - | 0 | - |
| 20 | 22 | - | 0* | - |

*The non-lifted formulation cannot be generated on our test machines.

As suggested in Proposition 1, the convex hull for a ReLU node involves an exponential (in terms of number of inputs) number of non-lifted constraints:

$$y \leq \sum_{i \in \mathcal{I}_j} w_i x_i + \sigma(b + \sum_{i \in \mathcal{I} \setminus \mathcal{I}_j} UB_i) + (\sigma - 1)(\sum_{i \in \mathcal{I}_j} LB_i), \qquad \forall j = 1, ..., 2^\eta \qquad (68)$$

where $UB_i$ and $LB_i$ denote the upper and lower bounds of $w_i x_i$. The set $\mathcal{I}$ denotes the input indices $\{1, ..., \eta\}$, and the subset $\mathcal{I}_j$ contains the union of the $j$-th combination of partitions $\{\mathbb{S}_1, ..., \mathbb{S}_\eta\}$.

Anderson et al. [3] provide a linear-time method for identifying the most-violated constraint in (68). First, the subset $\hat{\mathcal{I}}$ is defined:

$$\hat{\mathcal{I}} = \left\{ i \in \mathcal{I} \mid w_i x_i < w_i \left( LB_{x_i}(1 - \sigma) + UB_{x_i}\sigma \right) \right\} \qquad (69)$$

where $LB_{x_i}$ and $UB_{x_i}$ are, respectively, the lower and upper bounds of input $x_i$. Then, the constraint in (68) corresponding to the subset $\hat{\mathcal{I}}$ is checked:

$$y \leq \sum_{i \in \hat{\mathcal{I}}} w_i x_i + \sigma(b + \sum_{i \in \mathcal{I} \setminus \hat{\mathcal{I}}} UB_i) + (\sigma - 1)(\sum_{i \in \hat{\mathcal{I}}} LB_i) \qquad (70)$$

If this constraint (70) is violated, then it is the most violated in the family (68). If it is not, then no constraints in (68) are violated. This method can be used to solve the separation problem and generate cutting planes during optimization.

Figure 2 (left) shows that dynamically generated, convex-hull-based cuts only improve optimization performance when added at relatively low frequencies, if at all. Alternatively, De Palma et al. [13] only add the most violated cut (if one exists) at the initial LP solution to each node in the NN. Adding the cut before solving the MILP may produce different performance than dynamic cut generation: the De Palma et al. [13] approach to adding cuts includes the cut in the solver (Gurobi) pre-processing. Tables 5–8 give the results of the optimal adversary problem (cf. Table 1) with these single "most-violated" cuts added to each node. These cuts sometimes improve the big-M performance, but partition-based formulations still consistently perform best.

Table 5: Number of solved (in 3600s) optimal adversary problems for big-M vs $N = \{2, 4\}$ equal-size partitions. Columns marked "w/cuts" denote most violated convex-hull cut at root MILP node added to each NN node. Most solved for each set of problems is in bold. Green text indicates more solved compared to no convex-hull cuts.

| Dataset | Model | $\epsilon_1$ | Big-M | | 2 Partitions | | 4 Partitions | |
|---|---|---|---|---|---|---|---|---|
| | | | w/ cuts | w/o cuts | w/ cuts | w/o cuts | w/ cuts | w/o cuts |
| MNIST | $2 \times 50$ | 5 | **100** | **100** | **100** | **100** | **100** | **100** |
| | $2 \times 50$ | 10 | **98** | 97 | **98** | **98** | **98** | **98** |
| | $2 \times 100$ | 2.5 | 95 | 92 | **100** | **100** | 97 | 94 |
| | $2 \times 100$ | 5 | 51 | 38 | 55 | **59** | 45 | 48 |
| | CNN1* | 0.25 | **91** | 68 | 85 | 86 | 83 | 87 |
| | CNN1* | 0.5 | 7 | 2 | **18** | 16 | 9 | 11 |
| CIFAR-10 | $2 \times 100$ | 5 | 45 | 62 | 65 | 69 | 61 | **85** |
| | $2 \times 100$ | 10 | 11 | 23 | 30 | 28 | 26 | **34** |

*OBBT performed on all NN nodes

Table 6: Average solve times of optimal adversary problems for big-M vs $N = \{2, 4\}$ equal-size partitions. Columns marked "w/cuts" denote most violated convex-hull cut at root MILP node added to each NN node. Average times computed for problems solved by all 6 formulations. Lowest average time for each set of problems is in bold. Green text indicates lower time compared to no convex-hull cuts.

| Dataset | Model | $\epsilon_\infty$ | Big-M | | 2 Partitions | | 4 Partitions | |
|---|---|---|---|---|---|---|---|---|
| | | | w/ cuts | w/o cuts | w/ cuts | w/o cuts | w/ cuts | w/o cuts |
| MNIST | $2 \times 50$ | 5 | 45.3 | 57.6 | 60.3 | **42.7** | 86.0 | 83.9 |
| | $2 \times 50$ | 10 | 285.2 | 431.8 | 305.0 | **270.4** | 404.5 | 398.4 |
| | $2 \times 100$ | 2.5 | 505.2 | 525.2 | 309.3 | **285.1** | 597.4 | 553.9 |
| | $2 \times 100$ | 5 | 930.5 | 1586.0 | 691.7 | **536.9** | 1131.4 | 1040.3 |
| | CNN1* | 0.25 | **420.7** | 1067.9 | 567.2 | 609.4 | 784.6 | 817.1 |
| | CNN1* | 0.5 | 1318.2 | 2293.2 | 1586.7 | **1076.0** | 1100.9 | 2161.2 |
| CIFAR-10 | $2 \times 100$ | 5 | 1739.2 | 1914.6 | 595.1 | 1192.4 | 834.8 | **538.4** |
| | $2 \times 100$ | 10 | 1883.3 | 1908.9 | **1573.4** | 1621.4 | 1767.44 | 2094.5 |

*OBBT performed on all NN nodes

Table 7: Number of solved (in 3600s) verification problems for big-M vs $N = \{2, 4\}$ equal-size partitions. OBBT was performed for all partitions, and columns marked "w/cuts" denote most violated convex-hull cut at root MILP node added to each NN node.. Most solved for each set of problems is in bold. Green text indicates more solved compared to no convex-hull cuts.

| Dataset | Model | $\epsilon_\infty$ | Big-M | | 2 Partitions | | 4 Partitions | |
|---|---|---|---|---|---|---|---|---|
| | | | w/ cuts | w/o cuts | w/ cuts | w/o cuts | w/ cuts | w/o cuts |
| MNIST | CNN1 | 0.050 | 78 | 82 | 91 | **92** | 89 | 90 |
| | CNN1 | 0.075 | 25 | 30 | **53** | 52 | 45 | 42 |
| | CNN2 | 0.075 | 16 | 21 | 34 | **36** | 11 | 31 |
| | CNN2 | 0.100 | 1 | 1 | **5** | **5** | 0 | 5 |
| CIFAR-10 | CNN1 | 0.007 | 98 | 99 | **100** | **100** | 99 | 99 |
| | CNN1 | 0.010 | 80 | 98 | 94 | **100** | 89 | **100** |
| | CNN2 | 0.007 | 78 | 80 | 94 | **95** | 73 | 68 |
| | CNN2 | 0.010 | 29 | 40 | **74** | 72 | 34 | 35 |

Table 8: Average solve times of verification problems for big-M vs $N = \{2, 4\}$ equal-size partitions. OBBT was performed for all partitions, and columns marked "w/cuts" denote most violated convex-hull cut at root MILP node added to each NN node. Average times computed for problems solved by all 6 formulations. Lowest average time for each set of problems is in bold. Green text indicates lower time compared to no convex-hull cuts.

| Dataset | Model | $\epsilon_1$ | Big-M | | 2 Partitions | | 4 Partitions | |
|---|---|---|---|---|---|---|---|---|
| | | | w/ cuts | w/o cuts | w/ cuts | w/o cuts | w/ cuts | w/o cuts |
| MNIST | CNN1 | 0.050 | 133.7 | 92.2 | 20.4 | **19.3** | 37.2 | 33.3 |
| | CNN1 | 0.075 | 504.7 | 239.0 | 155.8 | **84.4** | 178.2 | 161.2 |
| | CNN2 | 0.075 | 637.8 | 414.8 | 101.9 | **95.4** | 901.3 | 175.1 |
| | CNN2 | 0.100 | - | - | - | - | - | - |
| CIFAR-10 | CNN1 | 0.007 | 102.5 | 74.7 | **18.7** | 21.4 | 30.6 | 38.1 |
| | CNN1 | 0.010 | 599.4 | 66.5 | 68.5 | **19.1** | 137.1 | 33.9 |
| | CNN2 | 0.007 | 308.3 | 262.7 | **69.5** | 74.5 | 380.9 | 812.2 |
| | CNN2 | 0.010 | 525.9 | 394.2 | **100.6** | 147.4 | 657.0 | 828.2 |

*OBBT performed on all NN nodes