# OpenReview forum: "Partition-Based Formulations for Mixed-Integer Optimization of Trained ReLU Neural Networks"
_NeurIPS.cc/2021/Conference — NeurIPS 2021 Poster_

### Official Review · Reviewer_1SxU · 2021-07-13

**Rating:** 7
**Confidence:** 4

**Summary:**

This paper presents a family of mixed-integer optimization formulations for trained neural networks (or, rather, individual neurons in the network), with primary application to verification and adversarial analysis/training. The family traces out a spectrum between a small, weak formulation (the big-M) and a larger, stronger formulation (the "convex hull"). The family is parameterized by a partition of the multivariate input to a single ReLU neuron. A handful of strategies for doing this partition are presented, along with computational experiments.

**Ethical Concerns:**

No concerns.

**Limitations And Societal Impact:**

No concerns.

**Main Review:**

The main originality of the paper is the way in which existing formulation techniques can be combined to produce a family of approaches which attempt to balance between two important computational considerations. As such, there is not a ton of novelty, but the insight is a worthy contribution to the literature.

The paper is reasonably clear to the reader familiar with mixed integer optimization, though I do think that the presentation could be improved (suggestions to follow).

The computational results suggest that the contributions of the paper have merit and are a solid contribution to the verification literature. I do wish the authors left us with a clearer story on how N should be selected (either theoretically or practically, or both), but that may be too much to ask for a single conference paper.

Other comments:
* L14: The acronym "NN" is used without definition.
* L76: What is the "second equation" being referred to?
* L130: The description of the indexing of (20) is confusing. Should I understand that: you consider each nonempty subset of elements of {S_1,...,S_N}, assign it an integer index j, and then construct I_j as the union over all selected elements?
* Proposition 1 looks strikingly similar to the ideal formulation of Anderson et al. In particular, if we take each S_i = {i}, are they not the same, modulo how the LBs and UBs are computed? This connection should be made explicit.
* L147: Broken references
* The typesetting for (22), and similar blocks below, are very confusing. They read like a disjunction, with the "or", but the text uses them in a very different way. Consider other ways to typeset this discussion.
* The discussion of partitioning strategies is brief, with no real attempt at a theoretical justification for the chosen approaches. However, I don't find this a major deficiency. I do wish that the discussion, particularly for the "equal-range partitions", was written more clearly; it is sufficiently hard to parse that I do not believe that I could recreate the authors approach solely from the description in the text.
* Is the time reported in the tables accounting for the time spent on OBBT, or solely for the MIP?
* How does the performance look for the new approaches _without_ OBBT? Put another way: how strongly dependent are the positive computational results on OBBT?
* The values for epsilon reported (5 in Tables 2 and 3) seem quite high. Are the inputs normalized? Does it vary for the differing input sizes of MNIST and CIFAR10?
* Figure 3: Please do not mix two charts with differing y-axes into one! It is very confusing and/or misleading to the reader.
* In the computational results: Is the "average" arithmetic of geometric mean? How do time outs factor into the average time columns? This is a situation where time limits can introduce weird artifacts. For example, method 1 might be very fast on half of the instances and completely unsolvable on the remaining, while method 2 is consistently solvable within the time limit, but slow. Depending on how the time limit is set, the average time of method 1 (computed with time outs included) can be made arbitrarily large.



**Time Spent Reviewing:**

2

---

> ### Author Response · Authors · 2021-08-09
> **Response to Reviewer 1SxU**
>
> Thank you for the insightful review and suggestions.  Please find our responses to the specific comments below:
>
> &nbsp;&nbsp;&nbsp;Bullet 1. We will clarify this at first use.
> &nbsp;&nbsp;&nbsp;Bullet 2. We will remove this remark, as multiple reviewers found it unclear.
> &nbsp;&nbsp;&nbsp;Bullet 3: The reviewer is correct; we will try to further clarify the presentation of this formulation. We believe that making the connection to Anderson et al. (as suggested in Bullet 4) will also improve its clarity.
> &nbsp;&nbsp;&nbsp;Bullet 4: The formulations are indeed equivalent at the extreme case of S_i = {i}. We will make this connection explicit, as also suggested by Reviewer NoD7.
> &nbsp;&nbsp;&nbsp;Bullet 5: We will fix the reference.
> &nbsp;&nbsp;&nbsp;Bullet 6: Thanks for the helpful observation. We will use “vs.” instead of “or” to avoid confusing these alternatives with mathematical disjunctions.
> &nbsp;&nbsp;&nbsp;Bullet 7: We will try to make the descriptions of partitioning methods clearer, but we note that the code is also provided to help reproduce the results.
> &nbsp;&nbsp;&nbsp;Bullet 8: The time reported in tables solely accounts for the MIP; this will be clarified in the updated paper. We omit the time spent on OBBT in the tables for two reasons: all methods are tested withand benefit from OBBT, and OBBT problems for each layer are independent and can be solved in parallel. As described in lines 267–271, this may render the number of OBBT problems per layer of lesser importance.
> &nbsp;&nbsp;&nbsp;Bullet 9: Figure 3 compares runs with OBBT (dashed) and without (solid). Computational results are indeed strongly dependent on performing OBBT, but our proposed formulations outperform the alternative formulations in either case.
> &nbsp;&nbsp;&nbsp;Bullet 10: The off-the-shelf MNIST and CIFAR-10 datasets were used without further pre-processing. We note that in this work we consider both the 1 and ∞ norms for input perturbations.  For the case of the 1-norm, an epsilon value of $\epsilon_1= 5$ is actually quite small. If the perturbation were spread across inputs evenly, $\epsilon_1 = 5$ would correspond to an ∞-norm epsilon of $\epsilon_\infty \approx 0.006$ and $\epsilon_\infty \approx 0.002$ for MNIST and CIFAR-10 models, respectively. On the other hand, the 1-norm allows individual inputs to be perturbed more, within the bounds of [0,1].
> &nbsp;&nbsp;&nbsp;Bullet 11: Thanks for this suggestion; we wanted to include as much information and results in the main text as possible. We will visually separate the two as much as possible, e.g., by making Figures 3a and 3b, adding a solid line between the graphs, and/or increasing the horizontal spacing.
> &nbsp;&nbsp;&nbsp;Bullet 12:  This is a good point about time limits.  We attempted to avoid “weird artifacts” from time limits by only computing average times for each set of problems on instances solved by all formulations; therefore, no time outs are included in the average times reported. This procedure is described in the captions of Tables 1–2. Because no time outs are included, we computed averages as the arithmetic mean. We will clarify these specifics in Appendix A.4: Experiment Details.

---

### Official Review · Reviewer_Nod7 · 2021-07-15

**Rating:** 7
**Confidence:** 4

**Summary:**

The paper introduces a class of novel mixed-integer programming formulations for optimizing the output of a trained ReLU network. Its approach is to model a ReLU as a disjunction where the preactivation function is partitioned in its indices. This results a parameterized formulation that can range from the traditional big-M formulation to an ideal formulation at its extremes. Since the former is weak but light and the latter is strong but large, this allows us to find a trade-off where the formulation is both reasonably strong and not too large so that its performance is better than either extreme. Moreover, the paper presents an optimization-based bounds tightening approach that can be applied to these formulation. The authors show that this formulation has good computational performance compared to its extremes for three classes of problems in the area of neural network verification.

**Limitations And Societal Impact:**

This is not discussed but also not particularly relevant for this work.

**Main Review:**

This paper offers a solid and novel contribution to the area of MIP formulations for trained neural networks. These problems are particularly relevant in the area of neural network verification, but they also appear when neural networks are used as surrogate models for optimization. The paper is overall well-written, cites relevant work, and easy to understand for a reader with the relevant MIP and disjunctive programming background (although it is not clear to me if this is an easy read for someone without this background).

There are two main methodological contributions here that are related but could be viewed as orthogonal: the development of a partition-based formulation for ReLUs by disaggregating disjunctions (along with partition heuristics), and the use of optimization-based bound tightening within ReLU formulations. Both ideas are very interesting and contribute to a practical advancement. In particular, it is known that OBBT bounds for the big-M values make a large difference in practice (e.g. [30] from paper, and this is also confirmed again in this paper in Figure 3), so it stands to reason that incorporating OBBT bounds into other inequalities of the ReLU results in computationally stronger results.

The computational setup appears sound to me and sufficiently detailed for publication. Three classes of problems are investigated, although they are very similar to each other. A comparison is made with both traditional big-M formulations used in various works, and with the closest related work (Anderson et al., [3] in the paper). The method proposed does appear to produce a significant improvement over big-M, and a slight improvement over an implementation of the cuts from Anderson et al. with optimally tuned cut frequency. Note that if it matched that work, the results would still be valuable because this implementation is simpler, and tuning the number of partitions would be easier than tuning the cut frequency in Anderson et al.

I recommend this paper for acceptance, but I propose a few minor corrections (particularly regarding mathematical formalization) and suggestions below.

Mathematical formalization issues:

1. The statements that (6)-(11) models (5), and (13)-(17) models (12) are not mathematically precise. Note that (5) (= (12)) models a disjunction of two unbounded sets, whereas the formulations model a disjunction of two bounded sets. What the formulations model are a bounded version of (5) and (12). In (12), the bounds are in the z variables and they can be different bounds for each side of the disjunction because they may depend on previous layers of the neural network, and one can take the disjunction into account to tighten them, which is what is done in Section 3.1. Some explanation along these lines should be added. I do like that (5) and (12) are cleanly stated and it is easier to understand without having to worry about bounds, so I would suggest clarifying and formalizing this right after they are defined, but independently of how you present it, this should be made mathematically precise.

2. Similarly to above, Proposition 3 requires bounding conditions for correctness, since (1) assumes that $x \in \mathbb{R}^\eta$ (and thus the convex hull of (1) would simply be the epigraph of the function and not the formulation in the paper).

3. The proof of Proposition 4 is slightly incomplete: it argues that it is strictly tighter because the constraints in (20) are facet-defining, but misses the possibility that two constraints are the same. However, it is easy to see that they are distinct because the support of all constraints are different. Adding "and distinct from each other" after facet-defining should be sufficient to fix this, since this is easy to see.

4. Line 86-87 states "these formulations are tighter than (6)-(11)". Does this mean (2)-(4), or do you mean they are tighter given the OBBT method (see note above about formalizing the disjunction)? This should be clarified. Bounds aside, they represent the exact same disjunction so there is no reason for them to be tighter than (6)-(11). Also, to be precise, you should say that the linear relaxations of these formulations are tighter.


Suggestions for improving connections to previous work:

5. In Proposition 1, I believe it is interesting to point out that the set of inequalities (20) is a particular subset of the inequalities (28a) in Proposition 12 in Anderson et al. where $UB_i$ and $LB_i$ would be given by interval arithmetic. This is completely expected given that we should recover the ideal formulation with $N = \eta$, but clarifying the exact connection between the two works is helpful for the reader to understand how this fits into the framework of Anderson et al. The strength of the partition-based method can be interpreted as preselecting a special subset of inequalities from the ideal formulation in Anderson et al. (although in this paper the lifted formulation is used).

6. Implicit in Proposition 1 is that OBBT can be applied directly to the cuts from Anderson et al. It would be nice to point this out explicitly; I even think that this counts as a contribution. In addition, I would be curious to know if you tried doing this. You can use the original separation method and then strengthen the cut with OBBT (it may no longer be the most violated cut, but that is fine). I do not think that this would be particularly necessary for the NeurIPS publication, but if you have some spare time to include this experiment (probably in the appendix), it would be a very nice improvement to your contribution to the community.

7. A work that may be relevant to cite is: Tjandraatmadja, C., Anderson, R., Huchette, J., Ma, W., Patel, K. and Vielma, J.P., "The convex relaxation barrier, revisited: Tightened single-neuron relaxations for neural network verification", since this work describes the convex hull of a ReLU for a box input without binary variables.


Writing issues and other notes:

8. Line 76 states "Note that the second equation is not necessary to model y". Can you clarify in the paper what this second equation refers to? I am confused because all four equalities and inequalities in (5) are needed to model y as a disjunction. Or do you mean (3)?

9. Line 79: $LB^a$, $UB^a$, $LB^b$, and $UB^b$ are not defined.

10. In (12), should $y \leq 0$ be $y \geq 0$?

11. Line 147 has a broken reference.

12. Line 147: Could you explicitly mention that the experiments in Section 4 are done with respect to the lifted formulation? In my first read, I incorrectly interpreted that the projected formulation is used for the computational results. I think the Appendix A.5 results are a minor contribution that could be highlighted a bit more in the main text, since the opposite is true for the full extended formulation given the large number of variables.

13. Line 186: Is this sentence "The partitions ... constraints:" meant to be here?

14. Line 277: Typo: "verfication".

15. I have some difficulty fully understanding the left plot in Figure 2 and it could use some more contextualization. Could you define what "cut frequency" is? Does a frequency of k means that cuts are generated every 1/k nodes? What is the baseline (i.e. no cuts, your version), and does a low cut frequency mean that cuts are still generated at the root? How many rounds of cuts are applied per node? Could you also clarify in the text that the setup is that you start with your partition-based formulation and apply cuts?

16. Could you clarify that Figure 2 is without OBBT, if that is the case?

17. In Figure 3, why is it that N = 1 shows different values for non-OBBT? Is this just MIP solver variability?

**Time Spent Reviewing:**

6

---

> ### Author Response · Authors · 2021-08-09
> **Response to Reviewer Nod7**
>
> We are grateful for the detailed assessment and the helpful suggestions. Please find our responses to specific points below.
>
> Mathematical formalization issues:
> &nbsp;&nbsp;&nbsp;1-2. Thank you for noting this important difference between disjunctions of unbounded and bounded sets.  We will correct this issue and clarify this distinction in the updated paper for formulations (6)-(11), (13)-(17), and Proposition 3.  Specifically, we will clarify that the formulations are only exact representations for the case where the original disjunction is bounded, and that (13)-(17) represents the convex hull for the case where the original disjunction has the same bounds.
> &nbsp;&nbsp;&nbsp;3. As suggested, we will add “and distinct from each other” to the text.
> &nbsp;&nbsp;&nbsp;4. We will clarify that it is indeed the linear relaxations of these formulations that are tighter. While (13)-(17) and (6)-(11) do represent the exact same disjunction, the linear relaxation of (13)-(17) is tighter when $n >1$ for the case where partitions are bounded (whether from OBBT or interval arithmetic). Investigating the corresponding non-lifted formulations for the bounded disjunctions shows that (13)-(17) includes additional facet-defining constraints compared to (6)-(11).
>
> Connections to previous work:
> &nbsp;&nbsp;&nbsp;5. This is a nice observation. We had indirectly mentioned the equivalence (in terms of relaxation strength) in lines 204-205, but we will explicitly make the connection in Proposition 1 for clarity.
> &nbsp;&nbsp;&nbsp;6. Thank you for the valuable suggestion. We are interested in this idea because performing OBBT on the bounds in added cuts could improve their efficacy significantly. We hope to investigate this combination, but an efficient implementation of this strategy is not trivial, as it would involve solving (a large number of) OBBT problems during each solver callback.  Note that the low optimal cut frequencies in Fig 2 suggest that adding (non-OBBT) cuts via solver callbacks can significantly slow optimization performance. Making cut strengthening computationally viable may require a technique to obtain improved bounds for the partitioned variables that is significantly faster than OBBT.
> &nbsp;&nbsp;&nbsp;7. Thanks for pointing us to this relevant reference, which we will use to further contextualize our contribution.
>
> Writing issues and other notes:
> &nbsp;&nbsp;&nbsp;8. We will remove this remark, as multiple reviewers found it unclear.
> &nbsp;&nbsp;&nbsp;9-11, 14. We will correct these writing issues.
> &nbsp;&nbsp;&nbsp;12, 16. We will add these clarifications.
> &nbsp;&nbsp;&nbsp;13. We will remove this extra sentence.
> &nbsp;&nbsp;&nbsp;15.  A “cut frequency” of k indeed refers to generating a cut at every 1/k MIP nodes. With this implementation, a low cut frequency does generate cuts at the root node (alternatively the cuts could be added at random MIP nodes with probability 1/k); however, baseline results with no cuts and single cut only can be found in Tables 1 and 4, respectively. The most violated cut is added to each neural network node at MIP nodes where cuts are added. We will make these clarifications in the updated paper.
> &nbsp;&nbsp;&nbsp;17. The most likely cause of this discrepancy is variability in computational performance. For the non-OBBT cases, relatively few problems were solved in 3600s, and many solved problems were completed close to 3600s. Therefore, some runs could likely terminate slightly under 3600s, yet not finish in 3600s when repeated. This is supported by the consistent performance for the cases with OBBT, where most problems are solved in well-under 3600s.

---

> > ### Comment · Reviewer_Nod7 · 2021-08-10
> > **Additional comments**
> >
> > Thank you for the responses. These address my concerns and I am happy with the state of the paper, assuming these improvements will be properly made. I only have two additional comments on items 4 and 15:
> >
> > 4\. Thanks for clarifying. I do request to make this statement precise and point out the conditions for this to be true, as it is subtle. Let me know if I am misinterpreting something, but if you are indeed comparing with (6)-(11) and not (2)-(4), the statement is conditional on the bounds and N (e.g. if N = 1 then you get (2)-(4)). In particular, in your response you mention that the relaxation is tighter even if using interval arithmetic, but I believe that this claim is *not* true in this case.
> >
> > Let's say the bounds from both (6)-(11) and (13)-(17) are computed via interval arithmetic, and assume the tightest $N = \eta$. In this case, (20) becomes the ReLU formulation from Anderson et al., which is the projection of (6)-(11) as shown in that work, and thus they are equally tight. As another way to view it, both formulations represent (a lifting of) the disjunction (5) plus bounds on the input variables, since any interval arithmetic bounds are implied by them, and thus they are equivalent in strength. In contrast, applying LP-based OBBT to the partitions imbues bounds to your disjunction that may not be implied by the input bounds or an LP-based bound on the full affine function. In any case, as you can see, there is some subtlety in this claim and it needs to be properly formalized. A suggestion is to write the statement in the beginning of Section 3 as "can be tighter" instead of "are tighter" and prove this as a Proposition in Section 3.2.
> >
> > 15\. Thank you for the clarification. Please include the baseline with no cuts to the plot as well (in whichever form you prefer), even if it is already present in Table 1.

---

> > > ### Author Response · Authors · 2021-08-16
> > > **Response to Additional Comments**
> > >
> > > Many thanks for the thoughtful comments.
> > >
> > > 4\. Your observation is correct, formulation (13)–(17) cannot be tighter than the convex hull formulation presented by Anderson et al. unless we use more “sophisticated” bounds than interval arithmetic bounds. We will clarify this in the text. We note that (6)-(11) is a lifted but equivalent version of the big-M formulation ($N=1$), c.f. the convex hull ($N=\eta$). The introduction of the aggregated input variable $z$ (equivalent to a single “partition”) weakens the formulation. Therefore, (6)-(11) is an alternative form of the big-M, written in a form similar to our partitioned formulation (13)–(17). By comparing (6)–(11) with (13)-(17), we can easily see that our formulation has an advantage over the big-M formulation, even with interval arithmetic bounds. For example, consider (6)–(11) for a node with inputs $x_1$ and $x_2$:
> > > >(eq1) $w_1 x_1 + w_2 x_2 - z^b + \sigma b \leq 0$ \
> > > >(eq2) $z^b + (1-\sigma) b \geq 0$ \
> > > >(eq3) $y = z^b + (1-\sigma) b$ \
> > > >(eq4) $\sigma LB^a \leq w_1 x_1 + w_2 x_2 - z^b \leq \sigma UB^a$ \
> > > >(eq5) $(1-\sigma) LB^b \leq z^b \leq (1-\sigma) UB^b$
> > >
> > > where $z^a$ is eliminated using $z^a + z^b = w_1 x_1 + w_2 x_2$. On the other hand, (13)–(17) for the case of $N = \eta = 2$ becomes:
> > > >(eq6) $w_1 x_1 + w_2 x_2 - z_1^b - z_2^b + \sigma b \leq 0$ \
> > > >(eq7) $z_1^b + z_2^b + (1-\sigma) b \geq 0$ \
> > > >(eq8) $y = z_1^b + z_2^b + (1-\sigma) b$ \
> > > >(eq9) $\sigma LB_1^a \leq w_1 x_1 - z_1^b \leq \sigma UB_1^a$ \
> > > >(eq10) $\sigma LB_2^a \leq w_2 x_2 - z_2^b \leq \sigma UB_2^a$ \
> > > >(eq11) $(1-\sigma) LB_1^b \leq z_1^b \leq (1-\sigma) UB_1^b$ \
> > > >(eq12) $(1-\sigma) LB_2^b \leq z_2^b \leq (1-\sigma) UB_2^b$
> > >
> > > It can be seen that the bounds in (eq4) represent a summation of (eq9)–(eq10), and (eq5) is similarly a summation of (eq11)–(eq12). When interval arithmetic bounds are used, the bounds are indeed "implied," e.g., $LB^a = LB_1^a + LB_2^a$. But, the latter formulation is stronger; satisfying (eq9)–(eq10) guarantees that (eq4) is satisfied, while the reverse is not true. This is also observed in Fig. 1, where formulation (6)–(11), equivalent to $N=1$, has weaker continuous relaxations for both interval arithmetic bounds (top row) and optimization-based bounds tightening (bottom row).
> > >
> > > The reviewer is correct that there is indeed subtlety in the original claim(s).
> > > As suggested, we will update Section 3 and better formalize/clarify the writing in Section 3.2.
> > >
> > > 15\. We will include the baseline results to Figure 2 to facilitate comparison, as suggested.

---

> > > > ### Comment · Reviewer_Nod7 · 2021-08-17
> > > > **Response to clarification**
> > > >
> > > > Ah, I see my misunderstanding now. The formulation (6)-(11) is *not* the Balas extended formulation of the disjunction (5). Typically you would duplicate all of the variables, but here you aggregate $w^T x$ into $z$. This makes sense now. Thank you for the clarification.

---

### Official Review · Reviewer_cKru · 2021-07-16

**Rating:** 7
**Confidence:** 1

**Summary:**

This work proposes a novel MILP formulation of ReLU networks that can control the size of the considered problem in a trade-off against the tightness of the relaxation. Theoretical analysis is provided along with experiments.

**Limitations And Societal Impact:**

Yes

**Main Review:**

The problem seems well motivated and while I am not familiar with the recent advances in MILP formulation of DNs the paper is well written, well motivated, and well explained. Additionally the experimental details are clear enough to allow reproduction.

Many ablation studies are provided to see at which point in this trade-off one is able to obtain best computation times and comparison with recent work seems thorough.

I did not find English issues.

**Time Spent Reviewing:**

5

---

> ### Author Response · Authors · 2021-08-09
> **Response to Reviewer cKru**
>
> Many thanks for the analysis and positive feedback. We are glad that our paper is “well written, well motivated, and well explained” for a reader who considers themselves less familiar with the area.

---

### Official Review · Reviewer_bEy7 · 2021-07-16

**Rating:** 7
**Confidence:** 4

**Summary:**

This paper proposes a mixed-integer programming formulation for trained ReLU neural networks, which will enable people to solve many related problems like optimal adversary, verification and minimally distorted adversary. The main advantage of this formulation comes from partitioning node inputs into a number of groups and use the disjunctive programming to form the convex hull. By selecting different partition size, one can recover the convex hull or the big-M formulation. For the purpose of balancing model size and tightness of the formulation, numerical experiments suggest that moderate size of partitions performs the best.

**Limitations And Societal Impact:**

Yes, the authors mentioned the problem of the poor selection of hyperparameter, and formulation sometimes does not integrate well with default cuts.

**Main Review:**

This paper is nicely written, the studied problem is interesting, and experiments are extensive and convincing. However, I do have a few comments, concerns and questions outlined as follows.

1. Many details and closely related results/reports are contained in the appendix. I think this paper might be a better fit for the full-length journal paper. (This would also give the authors more space for clarifications.)
2. The main novelty of this paper comes from transforming the original big-M formulation into disjunctive formulation, and then introduce new variables to partition the summation terms, which will allow us to obtain a more refined upper/lower bounds. To that regard, I think the novelty is not fundamental.
3. The format for the formulations in this paper (including the appendix) should be properly adjusted and aligned.
4. Line 89, page 3: "$\sum_{n}z_n, n=1, \ldots, N$" should be "$\sum_{n=1}^N z_n$".
5. $z_n = \sum_{S_n} w_i x_i$ should be $z_n = \sum_{i \in S_n} w_i x_i$ across the paper.
6. Second disjunction in equation (12): "$y \leq 0$" should be "$y \geq 0$".
7. Equation (18), (19): I think (18) should be $\sum_i \min(A, B) + b$, and (19) should be $\sum_i \max(C, D) + b$, where $A,B,C,D$ are those terms inside the brackets.
8. Line 147: what are (??)?
9. For those convex hull based cuts, in order to add them, how do you solve the corresponding separation problem?
10. Equation (52) in Appendix: here the subscript should be $I'_j$. Also, what is the reason for listing inequality (50) and (52) here?
11. Inequalities (65), (66) in Appendix: you should mention what are these constraints (65), (66) when $w_n=0$.


**Time Spent Reviewing:**

7

---

> ### Author Response · Authors · 2021-08-09
> **Response to Reviewer bEy7**
>
> Thank  you  for  the  thoughtful  and  constructive  review.   Please  find  our  responses  to  the  comments/concerns/questions below:
>
> &nbsp;&nbsp;&nbsp;1. We agree that we have provided significant supporting material in the appendices. We have aimed to highlight the main message in the paper, with supporting points in appendix. We will go through this again to check, and we will also improve referencing to the appendix results in the main text (as also suggested by Reviewer Nod7).
> &nbsp;&nbsp;&nbsp;2. We firstly note that the presentation of the big-M as a disjunctive program (and the corresponding formulation) was included solely to illustrate formulations for bounded disjunctions. This intermediate step helps introduce the new formulations, given by (13)–(17). The fundamental advancements novel to this work are (i) introducing the partitioned terms, as well as partitioning strategies, and (ii) the way optimization-based bounds tightening complements our new idea of partitioning the terms. The latter does indeed obtain “more refined upper/lower bounds,” but, more importantly, the proposed formulations enable us to derive and exploit new partitioned bounds. The partitioning strategies are also an important, and previously unexplored, aspect that greatly affects the computational performance.
> &nbsp;&nbsp;&nbsp;3-8. Thanks for these observations. We will update the writing and broken references as appropriate for the revised version.
> &nbsp;&nbsp;&nbsp;9. The separation problem is solved using the linear-time algorithm included in the cut-generation strategy of Anderson et al., as referenced in line 250. For completeness, we will mention this and will also add a brief description of the separation problem to the appendix.
> &nbsp;&nbsp;&nbsp;10.  We will correct this typo.  The inequalities resulting from (50) and (52) are indeed redundant, as stated in line 481.  They are included in the appendix for completeness of the Fourier-Motzkin elimination step.
> &nbsp;&nbsp;&nbsp;11. Thanks for this note, we will clarify this point: equations (65)–(66) are written with $x_n^a$ and $x_n^b$ isolated for ease of identifying the multiple-choice convex hull formulation. However, in practice the bounds can implemented without $w_n$ in the denominator, e.g., see (13)–(17). Moreover, we note that $w_n= 0$ means the input can simply be ignored when modeling the node.

---

> > ### Comment · Reviewer_bEy7 · 2021-08-31
> > **Update my rating**
> >
> > Thank you for the clarification. I think the bullet 2 comment is convincing enough for me to realize the novelty of this paper. For that reason I decide to raise my score from 6 to 7.

---

### Decision · Program_Chairs · 2021-09-27

**Decision:**

Accept (Poster)

**Comment:**

In this paper, the authors introduce novel mixed-integer programming formulations to optimize the output of a trained ReLU network. This is an important problem that arises, e.g., in verifying robustness of deep neural networks and adversarial training. The approach is based on partitioning node inputs into a number of groups and forming the convex hull over the partitions via disjunctive programming. In particular, this approach recovers the convex hull and Big-M formulations by choosing different partition sizes. The authors show that the proposed formulation provides computational advantages with respect to the baselines in neural network verification. The reviews all agreed that the paper introduces an interesting and effective strategy, only expressed minor concerns and suggestions, and recommended acceptance. Please take into account the updated reviews when preparing the final version to accommodate the requested changes. Thank you for your submission to NeurIPS.